# Levels of and changes in socioeconomic inequality in delivery care service: A decomposition analysis using Bangladesh Demographic Health Surveys

**Mohammad Habibullah Pulok[1], Gowokani Chijere Chirwa[2]\*, Jacob Novignon[3], Toshiaki Aizawa[4], Marshall Makate[5]**

**1** Geriatric Medicine Research, Nova Scotia Health Authority, Canada, Halifax, Nova Scotia, Canada, **2** Department of Economics, University of Malawi, Chancellor College, Zomba, Malawi, **3** Department of Economics, Kwame Nkrumah University of Science and Technology, Kumasi, Ghana, **4** Waseda Institute for Advanced Study, Waseda University, Tokyo, Japan, **5** Health Economics & Data Analytics, School of Public Health, Curtin University, Perth, Western Australia, Australia

\* gowokani@gmail.com, gchirwa@cc.ac.mw

**Data Availability Statement:** This study used data from Bangladesh Standard DHS, 2014 and 2011, which are publicly available on the open repository

## Abstract

### Background

Socioeconomic inequality in maternity care is well-evident in many developing countries including Bangladesh, but there is a paucity of research to examine the determinants of inequality and the changes in the factors of inequality over time. This study examines the factors accounting for the levels of and changes in wealth-related inequality in three outcomes of delivery care service: health facility delivery, skilled birth attendance, and C-section delivery in Bangladesh.

### Methods

This study uses from the Bangladesh Demographic and Health Survey of 2011 and 2014. We apply logistic regression models to examine the association between household wealth status and delivery care measures, controlling for a wide range of sociodemographic variables. The Erreygers normalised concentration index is used to measure the level of inequalities and decomposition method is applied to disentangle the determinants contributing to the levels of and changes in the observed inequalities.

### Results

We find a substantial inequality in delivery care service utilisation favouring woman from wealthier households. The extent of inequality increased in health facility delivery and C-section delivery in 2014 while increase in skilled birth attendance was not statistically significant. Wealth and education were the main factors explaining both the extent of and the increase in the degree of inequality between 2011 and 2014. Four or more antenatal care (ANC4+) visits accounted for about 8% to 14% of the observed inequality, but the contribution of ANC4+ visits declined in 2014.

at https://dhsprogram.com/data/available-datasets.cfm.

**Funding:** The author(s) received no specific funding for this work.

**Competing interests:** The authors have declared that no competing interests exist.

## Conclusion

This study reveals no progress in equity gain in the use of delivery care services in this decade compared to a declining trend in inequity in the last decade in Bangladesh. Policies need to focus on improving the provision of delivery care services among women from poorer socioeconomic groups. In addition, policy initiatives for promoting the completion of quality education are important to address the stalemate of equity gain in delivery care services in Bangladesh.

## Introduction

By 2030, the Sustainable Development Goals (SDGs) aims to reduce global maternal mortality to less than 70 deaths per 100,000 live births and the incidence of neonatal and infant deaths to as low as 12 and 25 deaths per every 1,000 live births, respectively [1]. Achieving such progress requires an unreserved commitment by governments worldwide, through the provision of high-quality maternal care that prioritises improved access to adequate antenatal care (ANC) and facility-based birth deliveries attended by qualified health professionals. Access to high-quality maternal delivery care potentially minimises the risk of having a stillbirth or child loss due to intrapartum-related complications by about 20% [2]. Moreover, skilled birth attendants have the capacity to conduct safe deliveries including ability to detect, address and refer any complications such as haemorrhage or sepsis which are known to kill mothers during and after childbirth [3]. Despite the well-known and apparent benefits of high-quality delivery care for both mother and child, there is persistent and large socioeconomic inequality in access to and use of delivery care services in many low-middle income countries [4, 5].

Bangladesh has remarkably improved maternal and neonatal health in the last decade [6]. However, there is well-documented socioeconomic inequality in various indicators of maternal healthcare services [7–10]. Socioeconomic inequality in delivery care service utilisation favouring better-off women remains a topical issue among research and public policy circles in Bangladesh [8]. Several studies reported that socioeconomic inequality in delivery care services has decreased in Bangladesh until 2011. For example, socioeconomic inequality in skilled birth attendant and institutional delivery declined over 1993/94-2011 [11, 12]. In addition, pro-rich inequality in Caesarean section (C-section) delivery lessened by about 25% between 2004 and 2014 [7]. Despite the observed declines of inequality in the uptake of delivery care services, the overall magnitude of such inequality remains high compared to that of ANC services [13]. There is also higher regional variation in socioeconomic inequalities in delivery care services compared to ANC services [8]. Furthermore, a recent study has projected that existing socioeconomic inequality in delivery care services is most likely to persist until 2030 [14].

Improving maternal healthcare coverage at the national level and closing the gap among different countries was a key target in the Millennium Development Goals (MDGs) while reducing socioeconomic inequality within country is considered fundamental to the SDGs [15]. The idea is that policy strategies targeted at narrowing the socioeconomic divide could help bridge the gap in utilisation of delivery care services. A clear understanding of the underlying drivers of the observed inequality in health facility delivery, skilled birth assistance, and C-section delivery is not only an important step in the design of effective policies to improve the health and wellbeing of Bangladeshi women and their children, but also critical towards

the monitoring and evaluation of progress towards goals 3.1 and 3.2 of the SDGs. Quantifying and explaining socioeconomic inequalities are essential for health planners to design policies that target specific sub-groups of the population where health resources are the most needed.

There are limited studies to explain the factors accounting for socioeconomic inequalities in the utilisation of delivery care services in Bangladesh [7, 16]. However, these studies did not consider several methodological issues associated with the measurement and decomposition of socioeconomic inequality in binary healthcare outcomes. For example, changes in the average level of C-section delivery was ignored when comparing socioeconomic inequality between 2004 and 2014 in the study by Khan et al., 2018 [7]. In addition, the decomposition method was used to explain the underlying factors that contribute to inequality, but statistical inference of the estimates from the decomposition analysis was not provided. In addition, there is no evidence to know how the role of contributing factors of inequality changes between two periods. Therefore, this study aims to address limitations of earlier studies by measuring and explaining wealth-related inequalities in health facility delivery, skilled birthattendance, and C-section delivery between 2011 and 2014. We also unravel the factors associated with the changes in inequalities. The findings of this study have important implications in terms of enlightening health planners, government, and other public health stakeholders interested in contributing towards the design and implementation of policies to alleviate socioeconomic inequalities in delivery care services in Bangladesh and other developing countries.

## Materials and methods

### Data and sample

The study is based on data from the two most recent rounds of the Bangladesh Demographic and Health Survey (BDHS), implemented in 2011 and 2014. The BDHS is a nationally representative cross-sectional survey conducted in every three years as a part of the global DHS programme since 1991/92. The BDHS employed a complex multistage sample design to collect data on maternal health and healthcare utilisation from women of reproductive age (15–49 years). The response rate of these surveys was about 98% [17]. We restrict our analysis to women who have had at least one live birth in the last three years preceding the surveys and only consider data related to the most recent delivery if there were multiple live births within the timeframe. After discarding the observations (around 2%) with missing information on selected variables, our final sample consists of 4638 and 4481 women in 2011 and 2014, respectively.

**Measures of delivery care service.** We analyse three dichotomous indicators of delivery care services utilisation: health facility delivery, skilled birth attendance, and C-section delivery. Health facility delivery is coded as 1 if a mother gave birth in a health facility (e.g. public hospital, district hospital, maternal and child welfare centre, upazila health complex, upazila health & family welfare centre, other public sector facility, community clinic, private hospital/clinic, NGO static clinic, other NGO facility etc.) and 0 if the birth occurred at home. Skilled birth attendance takes the value of 1 if the birth was assisted by a medically trained professional (e.g. qualified doctor, nurse, midwife, family welfare visitor and community skilled birth attendant) and 0 otherwise. Lastly, C-section delivery is a binary variable which is equal to 1 if the birth was a C-section delivery and 0 otherwise. Definitions and measurements of outcome variables used in this study were similar in both surveys.

### Determinants of delivery care services

This study follows previous literature from Bangladesh and other developing countries to select the determinants of delivery care service utilisation [13, 18–22]. These include women's

age at the time of survey, age at marriage, religion, number of children ever born (parity), history of pregnancy complications, at least four visits for antenatal care (ANC4+), exposure to mass media, involvement in microcredit and education. Education of husband is also included as the predictor of delivery care services [22]. Region and place of residence (rural/urban) are the geographic variables. Since BDHS does not collect information on household expenditure or income, the measure of socioeconomic status (SES) in this study is an asset-based wealth index [23].

The wealth index was constructed using Principal Components Analysis (PCA), which comes pre-calculated in the BDHS [24, 25]. The wealth index includes a range of assets or belongings owned by the households. Examples of the assets are radio/TV, refrigerators, farmland, farming animals, house construction materials, water, sanitation infrastructure etc [24]. PCA is a multivariate statistical method that is widely used as a data reduction technique [25]. This method creates uncorrelated components with each component consisting of a linear weighted combination of the original asset variables [25, 26]. The resulting components are arranged in such a way that the first principal component explains the largest variability in the data [25]. The asset score (continuous variable) is used to rank households from the lowest to the highest to compute our measures of SES-related inequality. Households are categorised into five wealth quintiles from the poorest (quintile 1) to the richest (quintile 5) [26, 27]. A detailed description of the methods underlying the construction of the asset-based index are described elsewhere [25, 28].

## Statistical analysis

This study uses logistic regression analysis to examine the association between the outcomes of delivery care service with demographic and socioeconomic variables. We measure and explain socioeconomic inequality in the utilisation of delivery care services using the concentration index (CI) and the decomposition method [29, 30]. We estimate the CI as follows:

$$\text{CI} = \frac{2}{\mu} cov(y_i, r_i) \tag{1}$$

where $y_i$ is indicator of delivery care use for individual $i$, $r_i$ is the fractional ranking of individuals according to wealth index and $\mu$ is the mean of $y_i$. The value of this index falls between −1 and 1 [30]. A negative CI indicates higher utilisation among the poor (pro-poor) while a positive value suggests greater utilisation among the rich (pro-rich). The higher the absolute value of the CI is, the greater the extent of inequality.

The range of the CI becomes smaller when the variable of interest is a binary indicator. This is because of the lower and the upper bounds of the CI depending on the mean of the outcome variable [31]. As a result, the change in socioeconomic inequality measured by the CI could be affected considerably if the mean of the variable of interest changes over time [32]. Therefore, we use the Erreygers Index (EI) to address this problem [32]. The EI is basically a normalised version of the CI as below:

$$EI = 4\mu CI \tag{2}$$

The interpretation of EI is like the standard CI. A positive EI shows the distribution of delivery care services favouring women from wealthier households i.e pro-rich inequality and vice versa. We then employ the decomposition technique to partition wealth-related inequalities in delivery care services. We assume that $y_i$, utilisation of delivery care services is modelled

by an additively separable linear function of $X_j$ (a vector of covariates) as shown below:

$$y_i = \alpha + \sum_{j=1}^{J} \beta_j X_{ji} + \varepsilon_i \tag{3}$$

Using Eq (3), the EI can be decomposed into the weighted sum of the socioeconomic inequality in the determinants for delivery care use [33]. Weight refers to the sensitivity of utilisation with respect to each covariate, which is defined by $\beta_j \bar{X}_j$. As our outcome variable is binary-thus providing more justification for the use of EI, the decomposition of the EI follows:

$$EI = 4\left[ \sum_{j=1}^{J} \beta_j \bar{X}_j * CI_j + GCI_\varepsilon \right] \tag{4}$$

In this expression, $\beta_j$ is the partial effects of healthcare determinants, $CI_j$ is the concentration indices of $X_j$ and $GCI_\varepsilon$ is the generalized CI of the error term. Eq 5 suggests that a variable contributes to inequality in the use of delivery care services when two conditions are satisfied: (1) it must be correlated with delivery use and (2) it must be unequally distributed across socioeconomic status as measured by the EI [23]. The higher is the partial effect of a variable and the more unequally the variable is distributed with respect to SES, the higher the contribution of that variable. We run linear probability models to estimate the coefficients of determinants of delivery care utilization in the decomposition analysis as a non-linear model could induce approximation error [34].

We are also interested in understanding how inequality has changed between 2011 and 2014 and the role of contributing factors in this change. For this purpose, we apply the Oaxaca-type decomposition following [33] to the EI. This procedure is shown in the following equation:

$$\Delta EI_t = \sum_{j=1}^{J} \beta_{jt} \bar{X}_{jt}(CI_{jt} - CI_{jt-1}) + \sum_{j=1}^{J} CI_{jt-1}(\beta_{jt}\bar{X}_{jt} - \beta_{jt-1}\bar{X}_{jt-1}) + \Delta(GCI_\varepsilon) \tag{5}$$

In Eq (5), $\Delta$ stands for differences across time. This method allows us to decompose the evolution of SES-related inequality in delivery care use into two components. The first component $((\sum_{j=1}^{J} \beta_{jt}\bar{X}_{jt}(CI_{jt} - CI_{jt-1}))$ represents the changes in socioeconomic inequality in the determinants while the second one $(\sum_{j=1}^{J} CI_{jt-1}(\beta_{jt}\bar{X}_{jt} - \beta_{jt-1}\bar{X}_{jt-1})$ measures the changes in the sensitivity of utilisation with respect to each covariate over time [29]. We follow the method of van Doorslaer et.al. [34] to obtain the standard errors of the CIs and their contributions by applying the bootstrapped method with 1,000 replications. This allows us to make the statistical inferences of the point estimates. We account for the multistage survey design in descriptive, regression, and decomposition analyses by using the sample weights, clusters, and strata provided in the BDHS data sets. All analyses are performed using Stata/MP version 15.1(Stata Corp., College Station, TX, USA).

## Ethics statement

The BDHSs 2011 and 2014 were implemented under the authority of the National Institute of Population Research and Training (NIPORT), the Ministry of Health and Family Welfare, Bangladesh. Mitra and Associates, a Bangladeshi research firm located in Dhaka conducted the surveys with technical assistance from the ICF International of Calverton, Maryland, USA. Institutional Review Board of the InnerCity Fund (ICF) Macro, Maryland, USA, and the National Research Ethics Committee of Bangladesh Medical Research Council (BMRC), Dhaka, Bangladesh approved the protocol of these surveys. Verbal consents were taken from

the participants before conducting in the interviews. We obtained the de-identified data for this study from the DHS online [35].

# Results

## Summary statistics

Table 1 reports that the percentage of women having skilled birth attendance increased from 31.9% in 2011 to 42.9% in 2014. The coverage of facility delivery increased from about 29% in 2011 to about 39% in 2014. There was an increase in C-section delivery by about 7 percentage point over the same period (17.2% in 2011 to 25.2% in 2014). About 27.1% of women were married by the age of 12–14 in 2014 compared to about 33.4% in 2011. The number of women with more than three children declined from 34.6% in 2011 to 30.0% in 2014. The coverage of at least four ANC visits increased from 25.5% in 2011 to 31.2% in 2014. The completion rate of secondary and higher education increased among both women and their husbands in 2014.

## Predictors of delivery care services

Regression results in Table 2 show that women having more than three children were significantly (OR: 0.40 with $p<0.01$) less likely to use all three services, compared with women with only one child. At least four ANC visits was positively associated with the likelihood of the uptake of delivery care services. For example, the likelihood of facility delivery for those with ANC4+ was about 2.4 ($p \leq 0.01$) times higher in 2011 and 1.9 ($p \leq 0.01$) times higher in 2014 than those who had less than four ANC visits. Educational attainment of both women and their husbands was significantly associated with the use of delivery care services. For example, women married to a husband with higher educational achievement were about 2.6 times more likely to have a C-section delivery, compared with women whose husbands had no primary education attainment in 2014. Women from poorer households were significantly less likely to use delivery care services compared to their counterparts from the richer household. The wealth gradient in the uptake of delivery care services became steeper in 2014 as shown by the higher OR of the richer quintiles. For example, the OR of giving birth at a health facility in the richest quintile increased from 3.19 in 2011 to 4.59 in 2014.

## Inequality in delivery care

There was an overall increase in the use of delivery care services across all wealth quintiles from 2011 to 2014, but the uptake of these services was lower among women from poorer households compared to women in wealthier households (Fig 1). It is noticeable that the absolute increase in the utilisation was higher among the women from wealthier households between 2011 and 2014. We also show this gradient in Fig 2 which plots the predicted probabilities from logistic regression models which include the interaction between year and wealth quintiles. The increase in predicted probabilities for facility delivery and C-section delivery were the highest and significant among women from the richest wealth quintile since there was no overlap in the 95% confidence intervals between 2011 and 2014. On the other hand, there was no notable change in the predicted probabilities between 2011 and 2014 for women from other wealth quintiles for these two outcomes.

Table 3 shows that the estimates of inequality were positive and statistically significant in both years. This result suggests that the distribution of delivery care services utilization was concentrated among women from wealthier households. There was an increase the value of the EI for all three outcomes. For example, the EI for facility delivery increased from 0.41 in 2011 to 0.47 in 2014 while that for C-section delivery increased from 0.31 in 2011 to 0.38 in

**Table 1. Weighted distribution of respondents by selected background characteristics.**

| Variables | 2011 | | 2014 | |
|---|---|---|---|---|
| | N | Proportion | N | Proportion |
| Facility delivery | 1350 | 29.1% | 1730 | 38.6% |
| Skilled birth attendance | 1480 | 31.9% | 1922 | 42.9% |
| C-section delivery | 798 | 17.2% | 1084 | 24.2% |
| Current age (Ref: 15–19) | 914 | 19.7% | 941 | 21.0% |
| 20–24 | 1739 | 37.5% | 1506 | 33.6% |
| 25–34 | 1725 | 37.2% | 1770 | 39.5% |
| 35+ | 260 | 5.6% | 269 | 6.0% |
| Age at marriage: Year 18+ | 1085 | 23.4% | 1268 | 28.3% |
| Year: 15–17 | 2004 | 43.2% | 1999 | 44.6% |
| Year: 12–14 | 1549 | 33.4% | 1214 | 27.1% |
| Parity (Ref: 1 child) | 1674 | 36.1% | 1788 | 39.9% |
| 2 children | 1359 | 29.3% | 1349 | 30.1% |
| 3 or more children | 1605 | 34.6% | 1344 | 30.0% |
| Religion (Ref: Islam) | 408 | 8.8% | 372 | 8.3% |
| Pregnancy complication (Ref: No) | 775 | 16.7% | 636 | 14.2% |
| ANC4+ visits (Ref: No) | 1183 | 25.5% | 1398 | 31.2% |
| Mass media exposure (Ref. No) | 1665 | 35.9% | 1716 | 38.3% |
| Irregular | 682 | 14.7% | 484 | 10.8% |
| Regular | 2291 | 49.4% | 2285 | 51.0% |
| Microcredit involvement (Ref: No) | 1498 | 32.3% | 1304 | 29.1% |
| Women education (Ref: No) | 821 | 17.7% | 636 | 14.2% |
| Primary | 1396 | 30.1% | 1250 | 27.9% |
| Secondary | 2078 | 44.8% | 2137 | 47.7% |
| Higher | 343 | 7.4% | 457 | 10.2% |
| Husband education (Ref: No) | 1285 | 27.7% | 1071 | 23.9% |
| Primary | 1382 | 29.8% | 1344 | 30.0% |
| Secondary | 1382 | 29.8% | 1420 | 31.7% |
| Higher | 589 | 12.7% | 645 | 14.4% |
| Wealth quintile (Ref. Poorest) | 1062 | 22.9% | 972 | 21.7% |
| Poorer | 918 | 19.8% | 851 | 19.0% |
| Middle | 914 | 19.7% | 856 | 19.1% |
| Richer | 904 | 19.5% | 923 | 20.6% |
| Richest | 835 | 18.0% | 883 | 19.7% |
| Urban resident (Ref: Rural) | 3576 | 77.1% | 3311 | 73.9% |
| Region (Ref: Barisal) | 260 | 5.6% | 260 | 5.8% |
| Chittagong | 1076 | 23.2% | 977 | 21.8% |
| Dhaka | 1415 | 30.5% | 1582 | 35.3% |
| Khulna | 441 | 9.5% | 358 | 8.0% |
| Rajshahi | 612 | 13.2% | 448 | 10.0% |
| Rangpur | 487 | 10.5% | 435 | 9.7% |
| Sylhet | 343 | 7.4% | 417 | 9.3% |

**Table 2. Multivariate logistic regression results for the factors associated with the use of delivery care services.**

| | Facility delivery | | | | | | Skilled birth attendance | | | | | | C-section delivery | | | | | |
|---|---|---|---|---|---|---|---|---|---|---|---|---|---|---|---|---|---|---|
| | 2011 | | | 2014 | | | 2011 | | | 2014 | | | 2011 | | | 2014 | | |
| | AOR | SE | P | AOR | SE | P | AOR | SE | P | AOR | SE | P | AOR | SE | P | AOR | SE | P |
| Age (years) at survey date (Ref: 15–19) | | | | | | | | | | | | | | | | | | |
| 20–24 | 1.04 | (0.16) | 0.81 | 1.16 | (0.16) | 0.27 | 0.99 | (0.15) | 0.95 | 1.03 | (0.14) | 0.81 | 1.51 | (0.26) | 0.02 | 1.09 | (0.17) | 0.60 |
| 25–34 | 1.48 | (0.28) | 0.04 | 1.66 | (0.30) | 0.01 | 1.42 | (0.25) | 0.05 | 1.49 | (0.28) | 0.04 | 2.30 | (0.49) | 0.00 | 1.71 | (0.34) | 0.01 |
| 35+ | 2.00 | (0.57) | 0.02 | 2.41 | (0.61) | 0.00 | 1.81 | (0.50) | 0.03 | 1.92 | (0.51) | 0.01 | 3.29 | (1.07) | 0.00 | 2.80 | (0.86) | 0.00 |
| Age (years) at marriage (Ref: 18+) | | | | | | | | | | | | | | | | | | |
| 15–17 | 0.89 | (0.12) | 0.37 | 0.77 | (0.09) | 0.02 | 0.95 | (0.12) | 0.70 | 0.79 | (0.08) | 0.02 | 1.02 | (0.14) | 0.89 | 0.66 | (0.10) | 0.01 |
| 12–14 | 0.69 | (0.11) | 0.03 | 0.70 | (0.11) | 0.02 | 0.69 | (0.11) | 0.02 | 0.69 | (0.10) | 0.01 | 0.78 | (0.15) | 0.18 | 0.73 | (0.14) | 0.10 |
| Parity (Ref: 1 child) | | | | | | | | | | | | | | | | | | |
| 2 children | 0.59 | (0.07) | 0.00 | 0.57 | (0.08) | 0.00 | 0.58 | (0.07) | 0.00 | 0.60 | (0.09) | 0.00 | 0.59 | (0.09) | 0.00 | 0.55 | (0.07) | 0.00 |
| 3 or more children | 0.40 | (0.07) | 0.00 | 0.41 | (0.06) | 0.00 | 0.42 | (0.07) | 0.00 | 0.41 | (0.07) | 0.00 | 0.36 | (0.08) | 0.00 | 0.39 | (0.08) | 0.00 |
| Religion (Ref: Islam) | 1.66 | (0.27) | 0.00 | 1.12 | (0.27) | 0.62 | 1.71 | (0.26) | 0.00 | 1.01 | (0.22) | 0.97 | 1.36 | (0.26) | 0.11 | 1.03 | (0.18) | 0.85 |
| Pregnancy complication (Ref: No) | 1.36 | (0.15) | 0.01 | 1.15 | (0.13) | 0.21 | 1.30 | (0.14) | 0.02 | 1.13 | (0.13) | 0.30 | 1.37 | (0.20) | 0.03 | 1.17 | (0.14) | 0.19 |
| ANC4+ visits (Ref: No) | 2.39 | (0.23) | 0.00 | 1.88 | (0.17) | 0.00 | 2.35 | (0.23) | 0.00 | 1.89 | (0.17) | 0.00 | 1.98 | (0.23) | 0.00 | 1.80 | (0.20) | 0.00 |
| Mass media exposure (Ref. No) | | | | | | | | | | | | | | | | | | |
| Irregular | 1.21 | (0.18) | 0.21 | 1.20 | (0.19) | 0.25 | 1.19 | (0.17) | 0.23 | 1.08 | (0.16) | 0.60 | 1.54 | (0.29) | 0.02 | 0.86 | (0.17) | 0.46 |
| Regular | 1.40 | (0.18) | 0.01 | 1.22 | (0.18) | 0.17 | 1.46 | (0.17) | 0.00 | 1.15 | (0.16) | 0.30 | 1.53 | (0.24) | 0.01 | 1.11 | (0.17) | 0.51 |
| Microcredit involvement (Ref: No) | 1.09 | (0.11) | 0.36 | 1.35 | (0.22) | 0.07 | 1.21 | (0.11) | 0.04 | 1.33 | (0.21) | 0.07 | 0.98 | (0.12) | 0.88 | 1.40 | (0.20) | 0.02 |
| Women education (Ref: No) | | | | | | | | | | | | | | | | | | |
| Primary | 1.15 | (0.21) | 0.43 | 1.54 | (0.27) | 0.01 | 1.10 | (0.18) | 0.56 | 1.57 | (0.25) | 0.00 | 1.14 | (0.27) | 0.58 | 1.40 | (0.29) | 0.11 |
| Secondary | 1.44 | (0.26) | 0.04 | 1.82 | (0.32) | 0.00 | 1.44 | (0.24) | 0.03 | 1.96 | (0.32) | 0.00 | 1.42 | (0.34) | 0.14 | 2.19 | (0.50) | 0.00 |
| Higher | 2.46 | (0.61) | 0.00 | 2.39 | (0.57) | 0.00 | 3.20 | (0.78) | 0.00 | 2.69 | (0.61) | 0.00 | 2.43 | (0.70) | 0.00 | 2.26 | (0.60) | 0.00 |
| Husband education (Ref: No) | | | | | | | | | | | | | | | | | | |
| Primary | 1.00 | (0.14) | 0.99 | 1.13 | (0.13) | 0.30 | 1.09 | (0.14) | 0.53 | 1.14 | (0.13) | 0.22 | 1.33 | (0.26) | 0.13 | 1.25 | (0.28) | 0.30 |
| Secondary | 1.32 | (0.19) | 0.05 | 1.34 | (0.19) | 0.04 | 1.41 | (0.19) | 0.01 | 1.45 | (0.20) | 0.01 | 1.77 | (0.33) | 0.00 | 1.65 | (0.39) | 0.03 |
| Higher | 1.85 | (0.33) | 0.00 | 2.10 | (0.39) | 0.00 | 1.80 | (0.32) | 0.00 | 2.60 | (0.48) | 0.00 | 2.54 | (0.56) | 0.00 | 2.58 | (0.71) | 0.00 |
| Wealth quintile (Ref. Poorest) | | | | | | | | | | | | | | | | | | |
| Poorer | 1.30 | (0.22) | 0.13 | 1.42 | (0.19) | 0.01 | 1.16 | (0.19) | 0.36 | 1.49 | (0.20) | 0.00 | 2.47 | (0.66) | 0.00 | 1.17 | (0.30) | 0.54 |
| Middle | 1.51 | (0.25) | 0.01 | 1.73 | (0.32) | 0.00 | 1.50 | (0.23) | 0.01 | 1.64 | (0.30) | 0.01 | 2.80 | (0.70) | 0.00 | 1.67 | (0.52) | 0.10 |
| Richer | 2.25 | (0.38) | 0.00 | 2.43 | (0.38) | 0.00 | 2.12 | (0.34) | 0.00 | 2.38 | (0.37) | 0.00 | 3.72 | (0.98) | 0.00 | 2.40 | (0.54) | 0.00 |
| Richest | 3.18 | (0.64) | 0.00 | 4.52 | (0.80) | 0.00 | 2.92 | (0.56) | 0.00 | 4.00 | (0.65) | 0.00 | 5.68 | (1.60) | 0.00 | 4.08 | (1.04) | 0.00 |
| Place of residence (Ref: Rural) | 0.68 | (0.08) | 0.00 | 0.71 | (0.09) | 0.01 | 0.62 | (0.07) | 0.00 | 0.77 | (0.09) | 0.03 | 1.18 | (0.16) | 0.22 | 0.84 | (0.11) | 0.19 |
| Region (Ref: Barisal) | | | | | | | | | | | | | | | | | | |
| Chittagong | 1.20 | (0.22) | 0.33 | 1.03 | (0.20) | 0.90 | 1.05 | (0.18) | 0.78 | 1.09 | (0.23) | 0.68 | 1.03 | (0.21) | 0.89 | 0.84 | (0.16) | 0.34 |
| Dhaka | 1.34 | (0.25) | 0.11 | 1.26 | (0.22) | 0.20 | 0.96 | (0.16) | 0.80 | 1.02 | (0.20) | 0.92 | 1.66 | (0.35) | 0.02 | 1.56 | (0.26) | 0.01 |
| Khulna | 2.72 | (0.50) | 0.00 | 2.61 | (0.54) | 0.00 | 2.04 | (0.35) | 0.00 | 2.13 | (0.48) | 0.00 | 2.03 | (0.43) | 0.00 | 2.03 | (0.36) | 0.00 |
| Rajshahi | 1.97 | (0.38) | 0.00 | 1.69 | (0.32) | 0.01 | 1.36 | (0.25) | 0.10 | 1.31 | (0.27) | 0.18 | 1.77 | (0.38) | 0.01 | 1.40 | (0.26) | 0.07 |
| Rangpur | 1.49 | (0.30) | 0.05 | 1.21 | (0.26) | 0.37 | 0.99 | (0.18) | 0.94 | 0.95 | (0.21) | 0.82 | 0.99 | (0.23) | 0.96 | 0.87 | (0.20) | 0.53 |
| Sylhet | 1.16 | (0.24) | 0.47 | 0.84 | (0.17) | 0.40 | 0.97 | (0.17) | 0.86 | 0.79 | (0.17) | 0.29 | 1.11 | (0.24) | 0.64 | 0.72 | (0.15) | 0.11 |

Notes: AOR = adjected odds ratio, SE = standard error, and robust standard errors in parentheses.

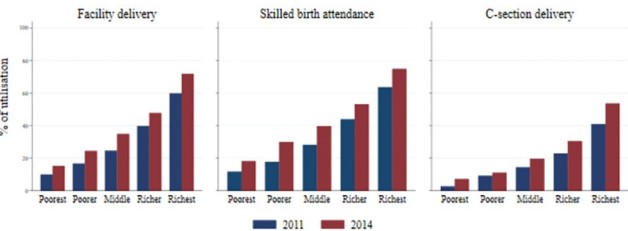

**Fig 1. Proportion of delivery care service utilisation by wealth quintiles.**

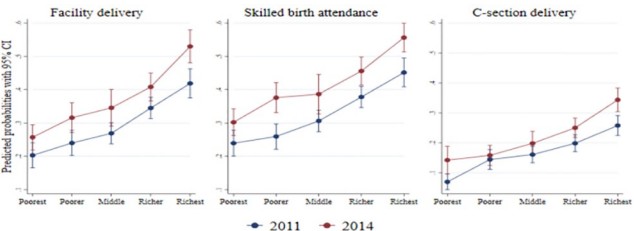

**Fig 2. Predicted probabilities of delivery care service utilisation across wealth quintile.**

2014. However, the increase in the EI for skilled birth attendance was not statistically significant.

## Decomposition of inequality in delivery care

The decomposition results in Table 4 suggest that household wealth explains about 36.3% of the socioeconomic inequality in facility delivery in 2011 and about 48.1% in 2014. For skilled birth attendance, its contribution increased from around 34.7% in 2011 to 45.0% in 2014. For C-section delivery, there was also an increase in the relative contribution of wealth status from about 38.4% in 2011 to 44.2% in 2014. The relative contribution of women's education reduced to 8.3% in 2014 from 10.8% in 2011 for facility delivery. On the other hand, husband's education explained about 10.4% of the inequality in skilled birth attendance in 2011 which increased to about 15.5% in 2014. ANC4+ visits had a positive contribution to the wealth-related inequality in delivery care, but its relative importance declined in 2014. For instance, its relative contribution to inequality in facility delivery was 14.0% in 2011, but it declined to about 8.8% in 2014.

**Table 3. Inequality in delivery care services in Bangladesh, 2011 and 2014.**

| | Facility delivery | | | | Skilled birth attendance | | | | C-section delivery | | | |
|---|---|---|---|---|---|---|---|---|---|---|---|---|
| | 2011 | | 2014 | | 2011 | | 2014 | | 2011 | | 2014 | |
| Inequality index | Estimate | SE | Estimate | SE | Estimate | SE | Estimate | SE | Estimate | SE | Estimate | SE |
| EI | 0.41*** | (0.02) | 0.47*** | (0.02) | 0.44*** | (0.02) | 0.47*** | (0.02) | 0.31*** | (0.02) | 0.38*** | (0.02) |
| Observations | 4638 | | 4481 | | 4638 | | 4481 | | 4638 | | 4481 | |

Notes: Standard errors in parentheses and significance level * p<0.10, ** p<0.05, *** p<0.01. EI = Erreygers corrected concentration index, SE = standard error.

**Table 4. Factor contributions to wealth-related inequalities in the use of delivery care services (decomposition of the EI).**

| | CI of factors | | Facility delivery | | | | Skilled birth attendance | | | | C-section delivery | | | |
|---|---|---|---|---|---|---|---|---|---|---|---|---|---|---|
| | 2011 | 2014 | 2011 | | 2014 | | 2011 | | 2014 | | 2011 | | 2014 | |
| Variables | CI | CI | Contr. | (%) | Contr. | (%) | Contr. | (%) | Contr. | (%) | Contr. | (%) | Contr. | (%) |
| Age (Ref: 15–19) | | | | | | | | | | | | | | |
| 20–24 | 0.000 | 0.041** | 0.000 | 0.000 | 0.002 | 0.391 | -0.000 | -0.001 | 0.000 | 0.087 | 0.000 | 0.003 | 0.001 | 0.340 |
| 25–34 | 0.033** | 0.004 | 0.003 | 0.774 | 0.001 | 0.127 | 0.003 | 0.683 | 0.000 | 0.086 | 0.005* | 1.722 | 0.000 | 0.131 |
| 35+ | -0.097* | -0.005 | -0.002 | -0.501 | -0.000 | -0.040 | -0.002 | -0.421 | -0.000 | -0.032 | -0.003* | -0.929 | -0.000 | -0.052 |
| Age at marriage (Ref: 18 +) | | | | | | | | | | | | | | |
| 15–17 | -0.014 | -0.038** | 0.001 | 0.129 | 0.003 | 0.693 | 0.000 | 0.071 | 0.003 | 0.645 | 0.000 | 0.012 | 0.004* | 1.163 |
| 12–14 | -0.150*** | -0.138*** | 0.011* | 2.708 | 0.010* | 2.083 | 0.012** | 2.733 | 0.010* | 2.182 | 0.004 | 1.407 | 0.008* | 1.983 |
| Parity (Ref: 1 child) | | | | | | | | | | | | | | |
| 2 children | 0.060*** | 0.040* | -0.006** | -1.531 | -0.005* | -1.064 | -0.007** | -1.523 | -0.005* | -0.989 | -0.005** | -1.558 | -0.004* | -1.174 |
| 3 or more children | -0.171*** | -0.177*** | 0.033*** | 7.939 | 0.034*** | 7.260 | 0.032*** | 7.408 | 0.035*** | 7.415 | 0.027*** | 9.001 | 0.028*** | 7.392 |
| Religion (Ref: Islam) | 0.060* | -0.049 | 0.002 | 0.453 | -0.000 | -0.072 | 0.002 | 0.457 | 0.000 | 0.004 | 0.001 | 0.287 | -0.000 | -0.050 |
| Pregnancy complication (Ref: No) | -0.006 | 0.029 | -0.000 | -0.042 | 0.000 | 0.081 | -0.000 | -0.035 | 0.000 | 0.067 | -0.000 | -0.043 | 0.000 | 0.074 |
| ANC4+ visits (Ref: No) | 0.329*** | 0.261*** | 0.058*** | 13.986 | 0.041*** | 8.750 | 0.058*** | 13.196 | 0.042*** | 8.904 | 0.035*** | 11.574 | 0.033*** | 8.650 |
| Mass media exposure (Ref. No) | | | | | | | | | | | | | | |
| Irregular | -0.165*** | -0.073* | -0.002 | -0.432 | -0.001 | -0.200 | -0.002 | -0.438 | -0.000 | -0.104 | -0.002 | -0.776 | 0.001 | 0.192 |
| Regular | 0.329*** | 0.336*** | 0.031** | 7.532 | 0.026 | 5.592 | 0.039*** | 8.969 | 0.021 | 4.435 | 0.022* | 7.129 | 0.009 | 2.465 |
| Microcredit involvement (Ref: No) | -0.148*** | -0.144*** | -0.003 | -0.653 | -0.009* | -1.877 | -0.006* | -1.300 | -0.009* | -1.881 | 0.000 | 0.014 | -0.007* | -1.847 |
| Women education (Ref: No) | | | | | | | | | | | | | | |
| Primary | -0.184*** | -0.256*** | -0.001 | -0.170 | -0.015* | -3.223 | 0.000 | 0.080 | -0.017* | -3.695 | 0.001 | 0.418 | -0.005 | -1.240 |
| Secondary | 0.189*** | 0.141*** | 0.014* | 3.413 | 0.024** | 5.062 | 0.016* | 3.682 | 0.030*** | 6.328 | 0.006 | 1.875 | 0.021** | 5.441 |
| Higher | 0.635*** | 0.553*** | 0.031*** | 7.531 | 0.030** | 6.505 | 0.038*** | 8.760 | 0.035*** | 7.366 | 0.033*** | 10.828 | 0.026** | 6.903 |
| Husband education (Ref: No) | | | | | | | | | | | | | | |
| Primary | -0.117*** | -0.171*** | 0.001 | 0.208 | -0.004 | -0.752 | -0.001 | -0.159 | -0.004 | -0.942 | -0.002 | -0.609 | -0.004 | -1.028 |
| Secondary | 0.233*** | 0.197*** | 0.012* | 2.933 | 0.015* | 3.116 | 0.016** | 3.559 | 0.021** | 4.408 | 0.014** | 4.458 | 0.016** | 4.125 |
| Higher | 0.545*** | 0.523*** | 0.034*** | 8.24 | 0.044*** | 9.413 | 0.031*** | 7.037 | 0.056*** | 12.026 | 0.039*** | 12.836 | 0.050*** | 13.326 |
| Wealth quintile (Ref. Poorest) | | | | | | | | | | | | | | |
| Poorer | -0.344*** | -0.377*** | -0.004 | -0.987 | -0.011 | -2.445 | -0.001 | -0.255 | -0.017* | -3.566 | -0.008* | -2.573 | 0.002 | 0.440 |
| Middle | 0.051*** | 0.003 | 0.001 | 0.334 | 0.000 | 0.044 | 0.002 | 0.382 | 0.000 | 0.038 | 0.002 | 0.493 | 0.000 | 0.016 |
| Richer | 0.443*** | 0.401*** | 0.039*** | 9.415 | 0.052*** | 11.166 | 0.040*** | 9.179 | 0.055*** | 11.779 | 0.026*** | 8.472 | 0.030** | 7.835 |
| Richest | 0.819*** | 0.803*** | 0.114*** | 27.565 | 0.184*** | 39.340 | 0.111*** | 25.382 | 0.172*** | 36.742 | 0.098*** | 32.036 | 0.136*** | 35.912 |
| Place of residence (Ref: Rural) | -0.142*** | -0.158*** | 0.029*** | 6.957 | 0.029** | 6.265 | 0.036*** | 8.309 | 0.022* | 4.744 | -0.008 | -2.532 | 0.014 | 3.806 |
| Region (Ref: Barisal) | | | | | | | | | | | | | | |
| Chittagong | 0.050*** | 0.099*** | 0.001 | 0.290 | 0.000 | 0.080 | 0.000 | 0.090 | 0.002 | 0.339 | 0.000 | 0.088 | -0.002 | -0.605 |
| Dhaka | 0.085*** | 0.112*** | 0.005* | 1.142 | 0.006 | 1.314 | -0.000 | -0.045 | 0.000 | 0.095 | 0.006** | 1.892 | 0.010* | 2.651 |
| Khulna | 0.087*** | -0.015 | 0.006*** | 1.338 | -0.001 | -0.188 | 0.004** | 0.983 | -0.001 | -0.158 | 0.003** | 0.935 | -0.001 | -0.145 |
| Rajshahi | -0.094*** | -0.138*** | -0.005** | -1.194 | -0.005** | -1.079 | -0.002 | -0.538 | -0.003 | -0.567 | -0.003* | -0.927 | -0.003 | -0.673 |
| Rangpur | -0.233*** | -0.210*** | -0.005* | -1.172 | -0.002 | -0.494 | 0.000 | 0.085 | 0.001 | 0.229 | 0.001 | 0.263 | 0.002 | 0.553 |
| Sylhet | -0.023 | -0.171*** | -0.000 | -0.038 | 0.001 | 0.245 | 0.000 | 0.005 | 0.002 | 0.444 | -0.000 | -0.031 | 0.001 | 0.357 |
| Explained inequality | | | 0.391 | 95.167 | 0.449 | 96.093 | 0.419 | 96.335 | 0.451 | 96.429 | 0.292 | 95.765 | 0.366 | 96.941 |

*(Continued)*

**Table 4.** (Continued)

| | CI of factors | | Facility delivery | | | | Skilled birth attendance | | | | C-section delivery | | | |
|---|---|---|---|---|---|---|---|---|---|---|---|---|---|---|
| | 2011 | 2014 | 2011 | | 2014 | | 2011 | | 2014 | | 2011 | | 2014 | |
| Variables | CI | CI | Contr. | (%) | Contr. | (%) | Contr. | (%) | Contr. | (%) | Contr. | (%) | Contr. | (%) |
| Residual | | | 0.016 | 4.833 | 0.018 | 3.907 | 0.016 | 3.665 | 0.017 | 3.571 | 0.013 | 4.235 | 0.012 | 3.059 |

Note: Significance level *** p<0.01, ** p<0.05, and * p<0.1. CI = concentration index, Contr = contribution (absolute).

## Decomposing changes in inequality in delivery care

Table 5 presents the results from the analysis of the decomposition of changes in inequality. Most of the change in inequality was associated with wealth status. The aggregate contribution (a sum of total changes for wealth variable) was 0.075 for facility delivery, 0.059 for skilled birth attendance, and 0.050 for C-Section. Notably, the large positive contributions were due to the sensitivity effects of the richest quintile. Their positive contributions were triggered by the steeper pro-rich gradient of delivery care service use in 2014. This result is clear from the finding that the coefficients of the richest quintile increased in 2014 for all the three outcomes. For example, the OLS coefficient estimate for facility delivery shows an increase from 0.188 to 0.272. On the other hand, the inequality effects of wealth variable show little contributions to the changes in the CIs of delivery care utilisation. In fact, the CI for the richest wealth quintile changed from 0.819 to 0.803 only. Husband's education was the second largest contributor to changes in inequality in facility delivery. The proportion of husbands who accomplished higher education increased from 12.7% to 14.4% between 2011 and 2014, and it was more concentrated among wealthier women in 2014.

## Discussion

The study measures and examines the extent of wealth-related inequalities in the utilisation of delivery care services in Bangladesh between 2011 and 2014. This study also explains the contributing factors that characterise the dynamics and the changes in the observed inequality. The findings reveal a substantial pro-rich inequality in the utilisation of three key components of delivery care services. Most importantly, the magnitude of absolute inequalities increased between 2011 and 2014 in health facility delivery and C-section delivery. Findings from the decomposition analysis indicate that household's wealth and education of both women and their husbands were the most important factors to explain the extent of and change in socioeconomic inequalities in delivery care in Bangladesh over the study period.

Our findings reveal a significant pro-wealth inequality in three outcomes of delivery care services, which is in line with earlier studies on socioeconomic inequalities in the use of maternal healthcare in Bangladesh [8, 9, 13, 36–38]. In a multi-country study, Bangladesh was ranked as the fourth most inequitable country in skilled birth attendance among 54 developing countries [39]. The extent of socioeconomic inequality in health facility delivery in Bangladesh is also one of the highest among the countries in South and East Asia [40]. We find that wealth-related absolute inequality in health facility delivery and C-section delivery increased between 2011 and 2014. This finding contradicts previous studies which measured socioeconomic inequality in maternal healthcare services in Bangladesh in the last two decades. For instance, a considerable reduction in inequality in delivery care service outcomes was documented between 1991 and 2011 [9]. A declining trend was also shown in other studies [13, 38]

**Table 5. Changes in the contributing factors of inequalities in delivery care service use (decomposition of the EI).**

| | Facility delivery | | | Skilled birth attendance | | | C-section delivery | | |
|---|---|---|---|---|---|---|---|---|---|
| | ΔSensitivity | ΔCI | Total | ΔSensitivity | ΔCI | Total | ΔSensitivity | ΔCI | Total |
| Current age (Ref: 15–19) | | | | | | | | | |
| 20–24 | 0.0000 | 0.0018 | 0.0018 | 0.0000 | 0.0004 | 0.0004 | 0.0000 | 0.0013 | 0.0013 |
| 25–34 | 0.0016 | -0.0042 | -0.0026 | 0.0007 | -0.0033 | -0.0026 | -0.0008 | -0.0040 | -0.0048 |
| 35+ | -0.0014 | 0.0032 | 0.0018 | -0.0007 | 0.0024 | 0.0017 | -0.0005 | 0.0031 | 0.0026 |
| Age at marriage: Year 18+ | | | | | | | | | |
| Year: 15–17 | 0.0006 | 0.0021 | 0.0027 | 0.0008 | 0.0020 | 0.0028 | 0.0015 | 0.0029 | 0.0044 |
| Year: 12–14 | -0.0005 | -0.0009 | -0.0014 | -0.0006 | -0.0011 | -0.0017 | 0.0040 | -0.0008 | 0.0032 |
| Parity (Ref: 1 child) | | | | | | | | | |
| 2 children | -0.0010 | 0.0024 | 0.0014 | 0.0000 | 0.0020 | 0.0020 | -0.0016 | 0.0020 | 0.0004 |
| 3 or more children | -0.0003 | 0.0013 | 0.0010 | 0.0009 | 0.0014 | 0.0023 | -0.0007 | 0.0011 | 0.0004 |
| Religion (Ref: Islam) | -0.0015 | -0.0008 | -0.0023 | -0.0021 | 0.0000 | -0.0021 | -0.0006 | -0.0004 | -0.0010 |
| Pregnancy complication (Ref: No) | 0.0001 | 0.0005 | 0.0006 | 0.0001 | 0.0004 | 0.0005 | 0.0001 | 0.0003 | 0.0004 |
| ANC4+ visits (Ref: No) | -0.0061 | -0.0105 | -0.0166 | -0.0052 | -0.0105 | -0.0157 | 0.0057 | -0.0083 | -0.0026 |
| Mass media exposure (Ref. No) | | | | | | | | | |
| Irregular | -0.0003 | 0.0012 | 0.0009 | 0.0008 | 0.0006 | 0.0014 | 0.0040 | -0.0009 | 0.0031 |
| Regular | -0.0058 | 0.0005 | -0.0053 | -0.0191 | 0.0004 | -0.0187 | -0.0126 | 0.0002 | -0.0124 |
| Microcredit involvement (Ref: No) | -0.0062 | 0.0002 | -0.0060 | -0.0034 | 0.0003 | -0.0031 | -0.0072 | 0.0002 | -0.0070 |
| Women education (Ref: No) | | | | | | | | | |
| Primary | -0.0101 | -0.0043 | -0.0144 | -0.0129 | -0.0048 | -0.0177 | -0.0047 | -0.0013 | -0.0060 |
| Secondary | 0.0177 | -0.0082 | 0.0095 | 0.0239 | -0.0104 | 0.0135 | 0.0221 | -0.0072 | 0.0149 |
| Higher | 0.0040 | -0.0045 | -0.0005 | 0.0016 | -0.0051 | -0.0035 | -0.0030 | -0.0039 | -0.0069 |
| Husband education (Ref: No) | | | | | | | | | |
| Primary | -0.0033 | -0.0011 | -0.0044 | -0.0023 | -0.0014 | -0.0037 | -0.0008 | -0.0012 | -0.0020 |
| Secondary | 0.0051 | -0.0027 | 0.0024 | 0.0088 | -0.0037 | 0.0051 | 0.0048 | -0.0028 | 0.0020 |
| Higher | 0.0117 | -0.0018 | 0.0099 | 0.0280 | -0.0024 | 0.0256 | 0.0134 | -0.0022 | 0.0112 |
| Wealth quintile (Ref. Poorest) | | | | | | | | | |
| Poorer | -0.0064 | -0.0010 | -0.0074 | -0.0142 | -0.0014 | -0.0156 | 0.0094 | 0.0001 | 0.0095 |
| Middle | 0.0018 | -0.0030 | -0.0012 | 0.0015 | -0.0030 | -0.0015 | -0.0004 | -0.0010 | -0.0014 |
| Richer | 0.0189 | -0.0057 | 0.0132 | 0.0211 | -0.0061 | 0.0150 | 0.0071 | -0.0033 | 0.0038 |
| Richest | 0.0736 | -0.0037 | 0.0699 | 0.0650 | -0.0036 | 0.0614 | 0.0409 | -0.0028 | 0.0381 |
| Place of residence (Ref: Rural) | -0.0026 | 0.0030 | 0.0004 | -0.0165 | 0.0023 | -0.0142 | 0.0206 | 0.0015 | 0.0221 |
| Region (Ref: Barisal) | | | | | | | | | |
| Chittagong | -0.0011 | 0.0002 | -0.0009 | 0.0004 | 0.0008 | 0.0012 | -0.0015 | -0.0011 | -0.0026 |
| Dhaka | -0.0002 | 0.0015 | 0.0013 | 0.0004 | 0.0001 | 0.0005 | 0.0018 | 0.0025 | 0.0043 |
| Khulna | -0.0005 | -0.0060 | -0.0065 | -0.0002 | -0.0049 | -0.0051 | 0.0002 | -0.0036 | -0.0034 |
| Rajshahi | 0.0016 | -0.0015 | 0.0001 | 0.0006 | -0.0008 | -0.0002 | 0.0011 | -0.0008 | 0.0003 |
| Rangpur | 0.0025 | 0.0002 | 0.0027 | 0.0010 | -0.0001 | 0.0009 | 0.0015 | -0.0002 | 0.0013 |
| Sylhet | 0.0003 | 0.0010 | 0.0013 | 0.0003 | 0.0018 | 0.0021 | 0.0003 | 0.0012 | 0.0015 |

in the last decade. However, these studies also acknowledged that the equity gain in delivery care services over time was not substantial compared to ANC services.

Our results show that inequality measured by the EI increased. This is because of an increase in the probability of delivery care services use between 2011 and 2014 among all the wealth quintiles (Figs 1 and 2). However, the absolute size of this improvement was larger among the richer women compared to their poorer counterparts. As a result, the percentage

point changes were greater among the richer women. Thus, it is expected that the EI estimates would increase, since it accounted for absolute changes in the outcome variable. On the other hand, the relative size of the improvement was larger among the poorer, which means that the percentage changes are greater among the poorer.

Previous studies reported that the absolute gap in the utilisation of delivery care services among different socioeconomic groups was widening over time despite an increase in utilisation rate among poorer women [10, 37]. For example. socioeconomic inequality in ANC4+ visits increased between 2011 and 2014 [10, 37]. A recent study projected that existing socioeconomic inequality in delivery care services is most likely to persist until 2030 [14]. This study further cautions that reaching the goal of 80% utilisation of maternal health care services by this time would not be possible despite substantial coverage of maternal health care interventions. In this regard, our findings add to the above concerns about no progress equity gain in delivery care services. However, further studies are required to affirm this conclusion for policymakers.

Our decomposition analysis reveals that wealth of the household and education were the most important factors that explain pro-rich inequality in delivery care services in Bangladesh, which is consistent with the current literature from different countries in South Asia, Middle East and sub-Saharan Africa [16, 41–45]. This result is attributable to the strong and positive association of these variables with delivery care service utilisation as well as pro-rich inequality in these variables. Women from richer households are more willing to pay for these services in the private sector, while poorer women may not be able to even bear the transportation cost to go to a public health facility [42]. The decomposition results also show that household's wealth status and husband's education contributed to the increase of socioeconomic inequality in facility delivery during the study period. These findings can be explained by the fact that the association of facility delivery with the women from the richest wealth quintile in 2014 became more string while there was almost no reduction in wealth inequality. Women from richer households were typically married with men with higher education, which contributed to the observed increase in inequality.

ANC visits also played a significant role in explaining the pro-rich inequality in delivery care utilisation, which is consistent with the findings from other studies [16]. However, the role of ANC4+ in explaining the socioeconomic inequality declined in 2014. Because the extent of association between visits to ANC four or more times and delivery care services declined during the period. Living in rural areas explained a significant contribution of pro-rich inequality in health facility delivery and skilled birth attendance as our regression results suggest that women in rural areas had lower utilisation of both services. This could be the result of the higher concentration of poorer women in rural areas and lower use of these services. Poor access to health facilities in rural areas makes the poorer women commute longer to get necessary care in the majority of the developing countries [42].

The interpretation and implications of the findings of this study are subject to a few limitations. For example, the role of supply-side factors in inequality of delivery care services have not been included in the models. This is due to the lack of information about accessibility and the quality of delivery care provision in the BDHS [20]. It could be the case that the utilisation of delivery care services was negatively influenced by greater distance to a health facility as reported in some other studies [46]. Recall bias could also induce measurement error in the outcome variables and our findings could be affected by this problem. However, childbirth is a very momentous life event for women and limiting the analysis to the latest birth would have potentially mitigated this limitation [8]. Another limitation is that decomposition exercise does not provide any causal interpretations with the findings [47], and it is rather an accounting practice to understand the amount of contribution [48].

Our study has important implications for policy. Improving average utilisation is easier than socioeconomic targeting. The government of Bangladesh has taken several policy measures to improve access to maternal healthcare services for reducing maternal and child mortality. These policies have led to improved utilisation of ANC and institutional delivery care services over the last two decades [20]. However, there has been no improvement to reduce socioeconomic inequality in delivery care services in recent years. Therefore, policies should focus on improving the accessibility of maternal health services, especially among the socioeconomically disadvantaged women. Our study has shown that the education of both women and their husbands plays a critical role in explaining inequality in delivery care service utilisation. In this regard, we emphasise that policies for promoting the completion of quality education are important in addressing this growing inequality. Given that income inequality is growing in Bangladesh in the face of rapid economic development in recent times [49], undertaking a redistributive policy reform is imperative to ameliorate inequality maternity care.

## Conclusion

This study adds to the literature by presenting robust empirical evidence on socioeconomic inequalities in the utilisation of delivery care services in Bangladesh. Our findings show that absolute inequalities in health facility delivery and C-section delivery increased 2011 and 2014. Compared to the progress in the reduction in socioeconomic inequality delivery care services measured by relative inequality indicator in the last decade, this study finds an increasing inequality measured by absolute inequality indicator in this decade in Bangladesh. Therefore, we emphasise to measure socioeconomic inequality using a robust indicator to present comprehensive evidence to policy makers. Our findings from this paper reinforce that policies need to focus on improving the provision of delivery care services among women from poorer socioeconomic groups. In addition, policy initiatives for promoting the completion of quality education are important to address the stalemate equity gain in the utilization of maternal healthcare services in Bangladesh.

## Acknowledgments

We are grateful to Professor Owen O'Donnell from Erasmus School of Economics, Erasmus University Rotterdam, for his insightful comments and suggestions on the earlier version of this paper.

## Author Contributions

**Conceptualization:** Mohammad Habibullah Pulok.

**Formal analysis:** Mohammad Habibullah Pulok, Gowokani Chijere Chirwa, Toshiaki Aizawa.

**Investigation:** Mohammad Habibullah Pulok, Gowokani Chijere Chirwa.

**Methodology:** Mohammad Habibullah Pulok.

**Supervision:** Gowokani Chijere Chirwa, Jacob Novignon.

**Validation:** Jacob Novignon, Marshall Makate.

**Visualization:** Gowokani Chijere Chirwa, Toshiaki Aizawa, Marshall Makate.

**Writing – original draft:** Mohammad Habibullah Pulok, Gowokani Chijere Chirwa, Jacob Novignon, Toshiaki Aizawa, Marshall Makate.

**Writing – review & editing:** Mohammad Habibullah Pulok, Gowokani Chijere Chirwa, Jacob Novignon, Toshiaki Aizawa, Marshall Makate.

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
