## [Decision Letter · Decision Letter 0]

26 May 2020

PONE-D-19-33387

Levels of and changes in socioeconomic inequality in delivery care service in Bangladesh

PLOS ONE

Dear Dr. Chirwa,

Thank you for submitting your manuscript to PLOS ONE. After careful consideration, we feel that it has merit but does not fully meet PLOS ONE’s publication criteria as it currently stands. Therefore, we invite you to submit a revised version of the manuscript that addresses the points raised during the review process.

We look forward to receiving your revised manuscript.

Kind regards,

David Hotchkiss

Academic Editor

PLOS ONE

Journal Requirements:

2. In your Methods section, please provide additional information on how the wealth index was calculated."

3. Our internal editors have looked over your manuscript and determined that it is within the scope of our Health Inequities and Disparities Research Call for Papers. This collection of papers is headed by a team of Guest Editors for PLOS ONE: Clare Bambra, Hans Bosma, Diana Burgess, Joseph Telfair, Barbara Turner, and Jennie Popay. The Collection will encompass a diverse range of research articles on health inequities and disparities.  Additional information can be found on our announcement page: https://collections.plos.org/s/health-inequities

If you would like your manuscript to be considered for this collection, please let us know in your cover letter and we will ensure that your paper is treated as if you were responding to this call. If you would prefer to remove your manuscript from collection consideration, please specify this in the cover letter.

Additional Editor Comments (if provided):

Given that the purpose of the study is to investigate changes in inequality over time, I agree with the second reviewer's comment that the period from 2011 to 2014 is too short and his suggestion to include in your analysis other national surveys carried out prior to 2011.

Reviewers' comments:

Reviewer's Responses to Questions

**Comments to the Author**

1. Is the manuscript technically sound, and do the data support the conclusions?

Reviewer #1: Partly

Reviewer #2: Partly

2. Has the statistical analysis been performed appropriately and rigorously? 

Reviewer #1: Yes

Reviewer #2: I Don't Know

3. Have the authors made all data underlying the findings in their manuscript fully available?

Reviewer #1: Yes

Reviewer #2: No

4. Is the manuscript presented in an intelligible fashion and written in standard English?

Reviewer #1: Yes

Reviewer #2: No

5. Review Comments to the Author

Reviewer #1: Major points:

1. The article’s message hinges on the choice of index (particularly between EI and WI)- while EI shows an increase over the study period the other two indices show a decline. The authors defend EI with their argument but it will be a good idea to mention a critique to their argument, as suggested in the paper below:

Kjellssona & Gerdthama, 2011. Correcting the Concentration Index for Binary Variables. Retrieved from https://project.nek.lu.se/publications/workpap/papers/WP11_4.pdf on April 2, 2020

I have pasted excerpts of the relevant text from pages 19-20:

“If the localization of the threshold within the distribution of the latent variable (compare Figure 5) is due to either arbitrariness or cultural differences, we argue that level independence is a desirable property; given that the level of prevalence is due to an arbitrary threshold, it is sensible if the measured degree of inequalities is invariant to the level of prevalence. ….

By contrast, the threshold for a diagnosed medical condition, e.g. diabetes, is less subjective and there is less variation between contexts. The prevalence of diabetes can therefore be considered to be both accurate and interesting information, and thus an analysis of relative inequalities may be appropriate. In such a case, level independence is not necessarily desirable and, given the normative acknowledgment of the mirror relativity of W, the choice of index should depend on the preferred value judgment.

Accordingly, we can sum up the discussion in a two by two matrix (see Figure 6). If the threshold of the latent variable is arbitrary or subjective, i.e. if there is a risk of reporting heterogeneity, one should consider using E. In turn, if the threshold is objective, the choice ought to depend on the imposed value judgment.”

The three indicators of delivery care belong to an objective category. The two indices differ mainly on the property of level independence and the choice of index is based on the “preferred value judgement”. The latter should be clarified, in explicit terms, in the text.

2. Please clarify the age-distribution in the data-sets. The 2011 Bangladesh DHS surveyed women from ages 13 years onward while 2014 Bangladesh DHS surveyed women from ages 15 years onward. The correction will most likely change the estimates for 2011.

3. The discussion section lacks clarity and needs to be strengthened. A major point of the study was to use EI instead of CI and WI but the authors have not touched on this point at all in the discussion. It would have been a good idea to expand on how the difference in calculations gives more insight into the changes in inequality over time. For example, the major recipients of delivery care in 2014 must have been in the richest 50% population (denominator of EI) thereby showing an increased EI but not in the richest 10% population (denominator of WI) thereby showing a decreased WI.

The study shows an increase in EI over time but this is due to changes in sensitivity to certain predictors and not due to changes in CI of the said predictors (which in fact show a decline). The authors fail to clearly demarcate this and explain or theorize on why this is so.

Additional points below.

Some minor points:

Lines 129-132: The description of health facility delivery: While there are no ‘health posts’ in the variable description of m15_1 in Bangladesh DHS 2011 or 2014, categories like upazila health & family welfare center; other public sector; community clinic; and other ngo sector have not been included in the description. Does it mean that these go into the “0” category?

Line 133: It will be a good idea to detail the categories included as ‘a medically trained professional’.

Line 247: Do the predicted probabilities refer to those obtained following the multivariate logistic regression? Please clarify.

Lines 249-251: That is a sweeping statement and does not apply to skilled care for the 2nd and 4th wealth quintile.

Lines 268-276: I will like to understand why the relative importance of 4+ ANC visits decreased. The same goes for the relative importance of household wealth, women’s education, and husband’s education. What does it signify? Were there some policy changes or some other ground realities that could be behind these findings?

Table 4 also raises the same questions as above. While we observe a reduction in inequality for independent factors (in 2nd & 3rd columns in Tables 4) like 4+ ANC visits, women’s and husband’s education (except for primary education), and in household wealth (except for the poorer), the relative contribution of these has changed in the models predicting the three delivery care indicators. Why? The explanation in “Lines 268-276” suggests substitution but again, the question is why?

Table 5: The above is further corroborated by the findings in Table 5. Here again, the findings suggest that the change in CI is mainly due to higher sensitivity to the parameters. On the other hand, the change in CI contributes negatively meaning that the direction of change for household wealth, women’s and husband’s education is the same as 4+ ANC visits. The discussion section should explain what this signifies.

Lines 324-325 & Lines 334-337: The authors refer to the same article for contradictory views, namely “reduction in inequality of delivery care services” and “increasing socioeconomic inequality”.

Lines 337-339: How did the authors reach this conclusion?

Lines 348-350: Please elaborate on this sentence.

Lines 353-354: 4+ ANC visits actually caused a reduction in CI from 2011 to 2014. When the authors say that it has contributed to inequality, I guess they refer to the positive CI in the 2nd & 3rd columns in Tables 4 (these are independent CI for each predictor for 2011 and 2014). This causes unnecessary confusion because so far, the narrative was about decomposition of inequality explained by a particular predictor within the multivariate model and its change over time.

Lines 387-388: How did the authors come to this conclusion? How and when was need defined?

Reviewer #2: This manuscript was previously submitted to other journal and unfortunately the authors did not follow the comments and suggestion that were made and they decided to submit the manuscript to PLOS ONE. The authors should consider reviewing the comments/suggestions from the reviewers before sending the manuscript to other journal.

6. PLOS authors have the option to publish the peer review history of their article (what does this mean?). If published, this will include your full peer review and any attached files.

Reviewer #1: Yes: Deepali Godha

Reviewer #2: No

---

## [Author Response · Author response to Decision Letter 0]

8 Aug 2020

we have attached the response document

---

## [Decision Letter · Decision Letter 1]

4 Sep 2020

PONE-D-19-33387R1

Levels of and changes in socioeconomic inequality in delivery care service: A decomposition analysis using Bangladesh Demographic Health Surveys

PLOS ONE

Dear Dr. Chirwa,

Thank you for submitting your manuscript to PLOS ONE. After careful consideration, we feel that it has merit but does not fully meet PLOS ONE’s publication criteria as it currently stands. Therefore, we invite you to submit a revised version of the manuscript that addresses the points raised during the review process.

We look forward to receiving your revised manuscript.

Kind regards,

David Hotchkiss

Academic Editor

PLOS ONE

Additional Editor Comments (if provided):

The manuscript has been re-reviewed by one of the reviewers that commented on the first version. Note that one of her suggestions -- incorporating the 2018 DHS data into the analysis -- has since been withdrawn, as it was determined that the 2018 dataset is not yet publicly available. However, there are still a few improvements that need to be made to the manuscript before we issue a decision to accept it for publication. Please review the reviewer's suggestions to modify the methodology section to reflect that CI is no longer presented in the study -- as well as the other comments and suggestions. I look forward to receiving a revised version of the manuscript, along with point by point responses.

Reviewers' comments:

Reviewer's Responses to Questions

**Comments to the Author**

1. If the authors have adequately addressed your comments raised in a previous round of review and you feel that this manuscript is now acceptable for publication, you may indicate that here to bypass the “Comments to the Author” section, enter your conflict of interest statement in the “Confidential to Editor” section, and submit your "Accept" recommendation.

Reviewer #1: (No Response)

2. Is the manuscript technically sound, and do the data support the conclusions?

Reviewer #1: Yes

3. Has the statistical analysis been performed appropriately and rigorously? 

Reviewer #1: Yes

4. Have the authors made all data underlying the findings in their manuscript fully available?

Reviewer #1: Yes

5. Is the manuscript presented in an intelligible fashion and written in standard English?

Reviewer #1: Yes

6. Review Comments to the Author

Reviewer #1: The authors have done a good job in explaining some of their results. However, there is still quite a bit of confusion in the corrected version owing to the removal of CI results, taking a stand on EI as opposed to WI and justifying its use, and use of some incorrect words. Also, after reading the other reviewer’s and the editor’s comments, I will like to point out that the 2018 Bangladesh DHS is available. It seems odd that the authors have not used it.

- If authors have removed the CI, then why do they still describe it in Methods section (Line 173 onwards)? Given that the authors have removed all initial results on CI, it does not make sense to then provide a decomposition of CI. If it is decomposition of EI, then table headings need to be corrected.

- It seems that the authors do not think that WI is useful (Lines 189-190 & 198-199). If the authors’ preference is EI (Lines 199-200), the need of explaining and estimating WI seems superfluous. Accordingly, a lot of unnecessary technical detail in the methods section can be removed and an explanation about EI and why it has been chosen should be enough.

- The authors refer to WI as relative inequality in Lines 363-364 while they have earlier explained WI as “WI is neither an absolute nor a relative measure of inequality” (Lines 199-200).

- Should the word be “decomposed” in Lines 185, 188, and 197? These refer to the corrections made to CI for estimation of WI and EI.

- The sentence in Lines 453-455 needs to be rechecked. This is because the declining inequality trend in last decade was probably measured using CI which even now shows a similar trend (though results have been removed). Also, the authors should be careful in using the words equity and inequity as opposed to equality and inequality.

- I will suggest the use of ‘skilled birth attendance’ instead of ‘skilled delivery’.

7. PLOS authors have the option to publish the peer review history of their article (what does this mean?). If published, this will include your full peer review and any attached files.

Reviewer #1: No

---

## [Editor Report · Decision Letter 2]

2 Nov 2020

Levels of and changes in socioeconomic inequality in delivery care service : A decomposition analysis using Bangladesh Demographic Health Surveys

PONE-D-19-33387R2

Dear Dr. Chirwa,

We’re pleased to inform you that your manuscript has been judged scientifically suitable for publication and will be formally accepted for publication once it meets all outstanding technical requirements.

Kind regards,

David Hotchkiss

Academic Editor

PLOS ONE
---

## [Editor Report · Acceptance letter]

11 Nov 2020

PONE-D-19-33387R2 

Levels of and changes in socioeconomic inequality in delivery care service: A decomposition analysis using Bangladesh Demographic Health Surveys 

Dear Dr. Chirwa:

I'm pleased to inform you that your manuscript has been deemed suitable for publication in PLOS ONE. Congratulations! Your manuscript is now with our production department. 

Kind regards, 

on behalf of

Dr. David Hotchkiss 

Academic Editor

PLOS ONE